# Buckling Behavior of Sandwich Cylindrical Shells Covered by Functionally Graded Coatings with Clamped Boundary Conditions under Hydrostatic Pressure

**DOI:** 10.3390/ma15238680

**Published:** 2022-12-05

**Authors:** Abdullah H. Sofiyev, Nicholas Fantuzzi, Cengiz Ipek, Gülçin Tekin

**Affiliations:** 1Coordination of General Courses, Istanbul Ticaret University, Istanbul 34445, Turkey; 2Scientific Research Centers for Composition Materials of UNEC, Azerbaijan State Economic University, Baku 1001, Azerbaijan; 3Department of IT and Engineering, Odlar Yurdu University, Baku 1072, Azerbaijan; 4Department of Civil, Chemical, Environmental, and Materials Engineering, University Bologna, 40136 Bologna, Italy; 5Department of Civil Engineering, Engineering Faculty, Istanbul Medeniyet University, Istanbul 34700, Turkey; 6Department of Civil Engineering, Civil Engineering Faculty, Yildiz Technical University, Istanbul 34220, Turkey

**Keywords:** sandwich cylindrical shells, FG coatings, clamped edges, hydrostatic buckling pressure, first order shear deformation theory

## Abstract

The buckling behavior of sandwich shells with functionally graded (FG) coatings operating under different external pressures was generally investigated under simply supported boundary conditions. Since it is very difficult to determine the approximation functions satisfying clamped boundary conditions and to solve the basic equations analytically within the framework of first order shear deformation theory (FOST), the number of publications on this subject is very limited. An analytical solution to the buckling problem of FG-coated cylindrical shells under clamped boundary conditions subjected to uniform hydrostatic pressure within the FOST framework is presented for the first time. By mathematical modeling of the FG coatings, the constitutive relations and basic equations of sandwich cylindrical shells within the FOST framework are obtained. Analytical solutions of the basic equations in the framework of the Donnell shell theory, obtained using the Galerkin method, is carried out using new approximation functions that satisfy clamped boundary conditions. Finally, the influences of FG models and volume fractions on the hydrostatic buckling pressure within the FOST and classical shell theory (CT) frameworks are investigated in detail.

## 1. Introduction

The most important applications of sandwich composites are found in advanced technology industries such as the aviation, aerospace, automotive, railroad and marine industries, due to their high stiffness/weight and strength/weight ratios which increase the load carrying capacity of structures and improve their performance while consuming less energy. The main disadvantage of sandwich structures made of traditional composites is that delamination cannot be prevented due to the different material properties on the contact surfaces of the core and the coating [1,2]. The ability to prevent such disadvantages in the applications of sandwich structural elements has led materials scientists to seek the creation of a new generation composite materials. The development of new technologies such as structural optimization and additive manufacturing has made it possible to realize their applications as functionally classified materials and microelectromechanical systems. These developments allow one to take into account the material properties of structural elements and extend representation beyond geometry. such material compositions and microstructures make object heterogeneous. Heterogeneous objects are primarily classified as multi-material objects with different material regions and functional class (or gradient) materials (FGMs), a new class of composites with continuous material and geometric changes. Recent research has focused on the design and fabrication of FGMs rather than multi-material objects [2].

FGMs are one of the revolutionary technologies being developed in the 21st century, and they consist of composites of two or more components whose properties depend on the composition gradient of one or all of the components [3]. Various production methods, such as the vapor deposition technique, powder metallurgy (PM), the centrifuge method, the solid free form technique, the manufacturing method, etc., are used for the production of FGMs [4]. The excellent properties of FGMs have led to their use as coatings in many new industries, producing components for automobiles, aircraft, turbine rotors, flywheels, gears, nuclear reactors, biomedicines (implants, bones), etc. [5,6,7,8,9]. The application of FG coatings as structural elements in high-tech industries has led to the need to examine their thermal, mechanical, chemical and physical properties as well as their buckling and vibration properties. The first attempts to solve the eigen value problems of FG-coated constructions started in 2005, and their solutions were carried out within the framework of various theories and with different methods. Notable among these studies, Zenkour [10] performed the mathematical modeling of the properties of FG-coated sandwich plates and presented comprehensive analyses by making the first attempt to solve the buckling and free vibration problems for simply supported boundary conditions. Sofiyev [11] studied the stability behavior of shear deformable FG sandwich cylindrical shells with freely supported edges under axial loads. Dung et al. [12] examined the buckling properties of simply supported FG sandwich truncated conical shells reinforced by FGM stiffeners filled with elastic foundations. Alibeigloo [13] devised a thermo-elasticity solution for FG sandwich circular plates using the generalized differential quadrature method. Sofiyev [14] investigated the stability response of shear deformable FG-coated truncated conical shells with simply supported boundary conditions subjected to axial loads. Moita et al. [15] reported active–passive damping in FG sandwich plate/shell structural elements. Hao et al. [16] analyzed the stability behavior of geometric nonlinear FG sandwich shallow shells using a newly developed displacement field.

Nguyen et al. [17] studied the buckling behavior of functionally graded plates with stiffeners based on the third-order shear deformation theory. Karroubi and Irani-Rahaghi [18] analyzed the free vibration of rotating simply supported sandwich cylindrical shells with an FG core and two FG layers. Sofiyev [19] analyzed the buckling and vibration of coating–FGM–substrate conical shells under hydrostatic pressure with mixed boundary conditions. Garbowski et al. [20] presented the torsional and transversal stiffness of orthotropic sandwich panels. Karakoti et al. [21] presented the free vibration response of P–FGM and S–FGM sandwich shell panels under simply supported boundary conditions using the finite element method. Hung et al. [22] analyzed the nonlinear buckling behavior of spiral corrugated sandwich FGM cylindrical shells surrounded by an elastic medium. Burlayenko et al. [23] developed an efficient finite element model based on conventional shell elements available in ABAQUS software for numerical solutions to the problems of the free vibration of FGM monolayers and sandwich flat panels with simply supported boundary conditions. Zhang et al. [24] presented static and dynamic analyses of FGPM cylindrical shells with quadratic thermal gradient distribution. Thai et al. [25] examined the bending of symmetric sandwich FGM beams with shear connectors. Dung et al. [26] used the third-order shear deformation theory for modeling the static bending and dynamic responses of piezoelectric bidirectional functionally graded plates. Duc and Vuong [27] solved the nonlinear vibration problem of shear deformable FGM sandwich toroidal shell segments by using the Galerkin method and the Runge–Kutta method. Shinde and Sayyad [28] developed a new higher-order shear and normal deformation theory for the solution of the static and free vibration problems of simply supported FGM sandwich shells. Ramezani et al. [29] analyzed the nonlinear dynamics of FG/SMA/FG sandwich cylindrical shells using HSDT and semi ANS functions. Chaabani, et al. [30] investigated the buckling of porous FG sandwich plates subjected to various nonuniform compressions using a finite element model based on the high-order shear deformation theory. Alsebai et al. [31] presented the semi-analytical solution to the problem of the thermo-piezoelectric bending of FG porous plates reinforced with graphene platelets. Sofiyev and Fantuzzi [32] solved the stability and vibration problem of clamped cylindrical shells containing FG layers within ST under axial loads. Hu et al. [33] presented a new analytical solution to the problem of the free vibration of non-Lévy-type functionally graded doubly curved shallow shells.

In the studies reviewed above, solutions to the eigen value problem of FG-coated sandwich shells were usually obtained for simply supported boundary conditions. It is very difficult to determine the approximation functions that satisfy the clamped boundary conditions in the framework of shear deformation theory (ST). In addition to this main difficulty, deriving the basic equations in the framework of ST for FG-coated sandwich cylindrical shells under the effect of a hydrostatic pressure load presents an additional difficulty. For this reason, analytical investigations of the mechanical behavior of FG sandwich shells under clamped boundary conditions are very limited. To address this shortcoming, in this study, the modeling and solution of the buckling problem of cylindrical shells with an FGM coating and isotropic core under external pressures under clamped boundary conditions are presented.

The study is constructed as follows: after the introduction, the material and geometric model of the problem is presented in Section 2, the basic relations and basic equations are derived in Section 3, the approximation function and the solution are obtained in Section 4, and Section 5 includes comparisons and original analyses.

## 2. Material and Geometric Model of Problem

Figure 1 presents two sandwich cylindrical shells of length *L* and radius *r* covered with coatings of functionally graded material whose core consists of two different isotropic materials: (a) a ceramic-rich core and (b) a metal-rich core. We assumed that the FG sandwich cylindrical shell with clamped edges was subjected to hydrostatic pressure. The thickness of the FG coatings, $h_{coat}$, is equal with the thickness of the core, $h_{core}$, and the total thickness of the sandwich cylindrical shell is h, i.e., $h=2h_{F}+h_{core}$. The origin of the coordinate system ($Ox_{1}x_{2}x_{3}$) is located on the reference surface of the core at the left end of the sandwich cylinder, with the $x_{1}$-axis pointing along the length of the cylinder, the $x_{2}$-axis in the circular direction, and the $x_{3}$-axis in the perpendicular direction to the $x_{1}x_{2}$ surface towards the center of curvature. One of the advantages of FG coatings in the preparation of sandwich structural elements is the formation of one surface from the metal-rich and the ceramic-rich surface, and the continuous and smooth change in properties from one surface to the other. Since the material properties are almost the same on the contact surfaces of the coatings as in the core in the formation of the sandwich structural elements, this advantage ensures that the layers do not break from each other at different loadings. In FG_1_/C/FG_1_ sandwich cylinders, the core is ceramic rich and the material properties of the FG coatings continuously change from metal-rich surface to ceramic-rich surface in the thickness direction (Figure 1a). In FG_2_/M/FG_2_ sandwich cylinders, on the other hand, the core is metal rich, and the material properties of the FG coatings constantly change from ceramic-rich surface to metal-rich surface (Figure 1b).

## 3. Basic Relations and Equations

The volume fractions ($V_{c}^{\left(k\right)}(k=1,2,3)$) of the coatings and core are obtained from a simple mixing rule of materials and are expressed as follows [10,11,12]:(1)$$\begin{array}{l} V^{\left(1\right)}={\left(\frac{x_{3}+0.5h}{h_{2}+0.5h}\right)}^{d},\;x_{3}\in \left[-h/2,h_{2}\right);\;V^{\left(2\right)}=1,\;x_{3}\in \left[h_{2},h_{3}\right];\;\\ V^{\left(3\right)}={\left(\frac{x_{3}-0.5h}{h_{3}-0.5h}\right)}^{d},\;x_{3}\in \left(h_{3},h/2\right] \end{array}$$
where d is the power law index and dictates the property dispersion profile and $V_{c}^{\left(k\right)}+V_{m}^{\left(k\right)}=1$ in the FG coatings.

The Young’s modulus and Poisson’s ratio of the FG coatings are mathematically modeled as follows [10,11]:(2)$$\begin{array}{l} Y_{fg_{1}}^{\left(k\right)}({\bar{x}}_{3})=Y_{m}e^{V^{\left(k\right)}\ln(Y_{c}/Y_{m})},\;\nu_{fg_{1}}^{\left(k\right)}({\bar{x}}_{3})=\nu_{m}e^{V^{\left(k\right)}\ln(\nu_{c}/\nu_{m})}\\ Y_{fg_{2}}^{\left(k\right)}({\bar{x}}_{3})=Y_{c}e^{V^{\left(k\right)}\ln(Y_{m}/Y_{c})},\;\nu_{fg_{2}}^{\left(k\right)}({\bar{x}}_{3})=\nu_{c}e^{V^{\left(k\right)}\ln(\nu_{m}/\nu_{c})} \end{array}$$

The material properties of the sandwich shells covered by the coatings with ceramic-rich or metal-rich cores are expressed as [11]:(3)$$\left[Y({\bar{x}}_{3}),\nu ({\bar{x}}_{3})\right]=\left\{ \begin{array}{l} Y_{fg_{i}}^{\left(1\right)},\;\nu_{fg_{i}}^{\left(1\right)}\;at\;x_{3}\in \left[-0.5h,h_{2}\right)\\ Y_{core_{i}}^{\left(2\right)},\nu_{core_{i}}^{\left(2\right)}\;at\;x_{3}\in \left[h_{2},h_{3}\right]\;\\ Y_{fg_{i}}^{\left(3\right)},\;\nu_{fg_{i}}^{\left(3\right)}\;at\;x_{3}\in \left(h_{3},0.5h\right]\;(i=1,2) \end{array}\right.$$
where ${\bar{x}}_{3}=x_{3}/h$, $Y_{fg_{i}}^{\left(k\right)},\nu_{fg_{i}}^{\left(k\right)}$ and $Y_{core_{i}}^{\left(2\right)},\nu_{core_{i}}^{\left(2\right)}$ are the Young moduli and Poisson ratios of the FG_1_ and FG_2_ coatings and the ceramic-rich and metal-rich cores, respectively.

The variations in the dimensionless Young moduli of the sandwich cylinders covered by the FG_1_ and FG_2_ coatings with ceramic-rich and metal-rich cores are illustrated in Figure 2 and Figure 3, respectively. Similar graphs can be drawn for other mechanical properties of the FG-coated sandwich shells.

The constitutive relationships of the elastic and isotropic layers of the FG-coated sandwich cylinders based on the FOST can be written as [11]:(4)$$\left[\begin{array}{l} \tau_{11}^{\left(k\right)}\\ \tau_{22}^{\left(k\right)}\\ \tau_{12}^{\left(k\right)}\\ \tau_{13}^{\left(k\right)}\\ \tau_{23}^{\left(k\right)} \end{array}\right]=\left[\begin{array}{l} q_{11i}^{\left(k\right)}\;q_{12i}^{\left(k\right)}\;0\;0\;0\\ q_{12i}^{\left(k\right)}\;q_{11i}^{\left(k\right)}\;0\;0\;0\\ 0\;0\;q_{66i}^{\left(k\right)}\;0\;0\\ 0\;0\;0\;q_{55i}^{\left(k\right)}\;0\;\\ 0\;0\;0\;0\;q_{44i}^{\left(k\right)} \end{array}\right]\left[\begin{array}{l} e_{11}\\ e_{22}\\ \gamma_{12}\\ \gamma_{13}\\ \gamma_{23} \end{array}\right]$$
where $\tau_{ij}^{\left(k\right)}$ (i=1,2,j=1,2,3) and $e_{ii}(i=1,2),\;\gamma_{ij}(i=1,2,j=2,3)$ are the stress and strain components, respectively, and $q_{ij}^{\left(k\right)}(i,j=1,2,6)$ are the coefficients depending on the normalized thickness coordinate and are defined as:(5)$$\begin{array}{l} q_{11i}^{\left(k\right)}=\frac{Y_{fg_{i}}^{\left(k\right)}({\bar{x}}_{3})}{1-{\left(\nu_{fg_{i}}^{\left(k\right)}\right)}^{2}},\;q_{12i}^{\left(k\right)}=\nu_{F_{i}}^{\left(k\right)}q_{11i}^{\left(k\right)},\;q_{44i}^{\left(k\right)}=q_{55i}^{\left(k\right)}=q_{66i}^{\left(k\right)}=\frac{Y_{fg_{i}}^{\left(k\right)}{\bar{x}}_{3}}{2\left[1+\nu_{{}_{ig}}^{\left(k\right)}\right]},\;(k=1,3)\\ q_{11i}^{\left(2\right)}=\frac{Y_{i}^{\left(2\right)}}{1-{\left(\nu_{i}^{\left(2\right)}\right)}^{2}},\;q_{12i}^{\left(2\right)}=\nu_{i}^{\left(2\right)}q_{11i}^{\left(2\right)},\;q_{44i}^{\left(2\right)}=q_{55i}^{\left(2\right)}=q_{66i}^{\left(2\right)}=\frac{Y_{i}^{\left(2\right)}}{2\left[1+\nu_{i}^{\left(2\right)}\right]}. \end{array}$$

It is assumed that the transverse shear stresses proposed by Ambartsumian [34,35] for homogeneous structural members and generalized to FG structural members in this study vary as follows depending on the thickness coordinate [11,34,35]:(6)$$\tau_{13}^{\left(k\right)}=\frac{df_{1}^{\left(k\right)}(x_{3})}{dx_{3}}\phi_{1}(x_{1},x_{2}),\;\tau_{23}^{\left(k\right)}=\frac{df_{2}^{\left(k\right)}(x_{3})}{dx_{3}}\phi_{2}(x_{1},x_{2})$$

Since the expression (6) is taken into account in the fourth and fifth of the system of Equation (4), the following expressions are obtained for shear strains $\gamma_{13}$ and $\gamma_{23}$:(7)$$\gamma_{13}=\frac{1}{q_{55i}^{\left(k\right)}(x_{3})}\frac{df_{1}^{\left(k\right)}(x_{3})}{dx_{3}}\phi_{1}(x_{1},x_{2}),\;\gamma_{23}=\frac{1}{q_{44i}^{\left(k\right)}(x_{3})}\frac{df_{2}^{\left(k\right)}(x_{3})}{dx_{3}}\phi_{2}(x_{1},x_{2})$$

Considering the assumptions of the FOST, the following relations are used [34,35]:(8)$$\frac{\partial u_{x_{1}}}{\partial x_{3}}=-\frac{\partial w}{\partial x_{1}}+\gamma_{13},\;\frac{\partial u_{x_{2}}}{\partial x_{3}}=-\frac{\partial w}{\partial x_{2}}+\gamma_{23}$$

When Equation (8) is integrated with respect to $x_{3}$ in the interval (0,$x_{3}$), and when $x_{3}$ = 0, $u_{x_{1}}=u(x_{1},x_{2})$ and $u_{x_{2}}=v(x_{1},x_{2})$, the expressions of displacements of any point of the shell are obtained as follows:(9)$$\left[\begin{array}{l} u_{x_{1}}\\ u_{x_{2}} \end{array}\right]=\left[\begin{array}{l} u-x_{3}\frac{\partial w}{\partial x_{1}}+J_{1i}^{\left(k\right)}(x_{3})\phi_{1}\\ v-x_{3}\frac{\partial w}{\partial x_{2}}+J_{2i}^{\left(k\right)}(x_{3})\phi_{2} \end{array}\right]$$
where u and v are the displacements of the axial and circumferential directions on the mid-surface, respectively, w is the deflection, $\phi_{1}(x_{1},x_{2})$ and $\phi_{2}(x_{1},x_{2})$ are the transverse normal rotations about the $x_{2}$ and $x_{1}$ axes, respectively, and the following definitions apply:(10)$$J_{1i}^{\left(k\right)}=\int_{0}^{x_{3}}\frac{1}{q_{55i}^{\left(k\right)}(x_{3})}\frac{df_{1}^{\left(k\right)}(x_{3})}{dx_{3}}dx_{3},\;J_{2i}^{\left(k\right)}=\int_{0}^{x_{3}}\frac{1}{q_{44i}^{\left(k\right)}(x_{3})}\frac{df_{2}^{\left(k\right)}(x_{3})}{dx_{3}}dx_{3},\;(k=1,2,3)$$

The strain components ($e_{11},\;e_{22},\;\gamma_{12}$) with $u_{x_{1}},u_{x_{2}},w$ of any point of the cylindrical shell can be defined by the following relations [35]:(11)$$e_{11}=\frac{\partial u_{x_{1}}}{\partial x_{1}},\;e_{22}=\frac{\partial u_{x_{2}}}{\partial x_{2}}-\frac{w}{r},\;\gamma_{12}=\frac{\partial u_{x_{1}}}{\partial x_{2}}+\frac{\partial u_{x_{2}}}{\partial x_{1}}$$

Substituting the expression (9) for the displacements of $u_{x_{1}}$ and $u_{x_{2}}$ into Equation (11), the following relations are obtained:(12)$$\left[\begin{array}{l} e_{11}\\ e_{22}\\ \gamma_{12} \end{array}\right]=\left[\begin{array}{l} e_{11}^{0}-x_{3}\frac{\partial^{2}w}{\partial x_{1}{}^{2}}+J_{1}^{\left(k\right)}\frac{\partial \phi_{1}}{\partial x_{1}}\\ e_{22}^{0}-x_{3}\frac{\partial^{2}w}{\partial x_{2}{}^{2}}+J_{2}^{\left(k\right)}\frac{\partial \phi_{2}}{\partial x_{2}}\\ \gamma_{12}^{0}-2x_{3}\frac{\partial^{2}w}{\partial x_{1}\partial x_{2}}+J_{1}^{\left(k\right)}\frac{\partial \phi_{1}}{\partial x_{2}}+J_{2}^{\left(k\right)}\frac{\partial \phi_{2}}{\partial x_{1}} \end{array}\right]$$
where $\left(e_{11}^{0},e_{22}^{0},\gamma_{12}^{0}\right)$ are the strain components on the mid-surface and are defined as:(13)$$\left[e_{11}^{0},\;e_{22}^{0},\;\gamma_{12}^{0}\right]=\left[\frac{\partial u}{\partial x_{1}},\;\frac{\partial v}{\partial x_{2}}-\frac{w}{r},\;\frac{\partial u}{\partial x_{2}}+\frac{\partial v}{\partial x_{1}}\right]$$

The force and moment components $T_{ij},\;Q_{i}$ and $M_{ij}$ of the FG-coated cylindrical shells are derived from the following integrals [32,34,35,36]:(14)$$\left(T_{ij},M_{ij},Q_{i}\right)=\sum_{k=1}^{3}\int_{h_{k}}^{h_{k+1}}\left[\tau_{ij}^{\left(k\right)},x_{3}\tau_{ij}^{\left(k\right)},\tau_{i3}^{\left(k\right)}\right]dx_{3}(i,j=1,2)$$

The stress function Φ is related to the forces as [34,35,36]:(15)$$\left(T_{11},\;T_{22},\;T_{12}\right)=h\left(\frac{\partial^{2}\Phi }{\partial x_{2}^{2}},\;\frac{\partial^{2}\Phi }{\partial x_{1}^{2}},\;-\frac{\partial^{2}\Phi }{\partial x_{1}\partial x_{2}}\right)$$

Taking the pre-buckling state of the sandwich cylinder for the membrane, the resultants $T_{11}^{0},\;T_{22}^{0},\;T_{12}^{0}$ are determined as [37]:(16)$$T_{11}^{0}=-Pr/2,\;T_{22}^{0}=-Pr,\;T_{12}^{0}=0$$

The stability and compatibility equations of the FG-coated cylindrical shells subjected to hydrostatic pressure are expressed as [36,37]:(17)$$\begin{array}{l} \frac{\partial M_{11}}{\partial x_{1}}+\frac{\partial M_{12}}{\partial x_{2}}-Q_{1}=0,\;\;\;\;\frac{\partial M_{12}}{\partial x_{1}}+\frac{\partial M_{22}}{\partial x_{2}}-Q_{2}=0,\\ \frac{\partial^{2}e_{11}^{0}}{\partial x_{2}^{2}}+\frac{\partial^{2}e_{22}^{0}}{\partial x_{1}^{2}}-\frac{\partial^{2}\gamma_{12}^{0}}{\partial x_{1}\partial x_{2}}+\frac{1}{r}\frac{\partial^{2}w}{\partial x_{1}^{2}}=0,\;\frac{\partial Q_{1}}{\partial x_{1}}+\frac{\partial Q_{2}}{\partial x_{2}}+\frac{T_{22}}{r}-\frac{rP}{2}\frac{\partial^{2}w}{\partial x_{1}^{2}}-rP\frac{\partial^{2}w}{\partial x_{2}^{2}}=0 \end{array}$$

By using the Equations (4), (12), (14)–(16) together, the expressions for the strains at the mid-surface, and forces and moments are obtained, and when the resulting expressions are substituted into the system of Equation (17), the basic equations of the FG-coated sandwich cylindrical shells subjected to hydrostatic pressure in the FOST framework take the following form:
(18)$$\begin{array}{l} L_{1}(\Phi ,w,\phi_{1},\phi_{2})\equiv \left(C_{1}-C_{5}\right)h\frac{\partial^{4}\Phi }{\partial x_{1}^{2}\partial x_{2}^{2}}+C_{2}h\frac{\partial^{4}\Phi }{\partial x_{1}^{4}}-C_{3}\frac{\partial^{4}w}{\partial x_{1}^{4}}-\left(C_{4}+C_{6}\right)\frac{\partial^{4}w}{\partial x_{1}^{2}\partial x_{2}^{2}}\\ +C_{7}\frac{\partial^{3}\phi_{1}}{\partial x_{1}^{3}}+C_{11}\frac{\partial^{3}\phi_{1}}{\partial x_{1}\partial x_{2}^{2}}-J_{3}\frac{\partial \phi_{1}}{\partial x_{1}}+(C_{8}+C_{12})\frac{\partial^{3}\phi_{2}}{\partial x_{1}^{2}\partial x_{2}}=0\\ L_{2}(\Phi ,w,\phi_{1},\phi_{2})\equiv C_{2}h\frac{\partial^{4}\Phi }{\partial x_{2}^{4}}+\left(C_{1}-C_{5}\right)h\frac{\partial^{4}\Phi }{\partial x_{1}^{2}\partial x_{2}^{2}}-\left(C_{6}+C_{4}\right)\frac{\partial^{4}w}{\partial x_{1}^{2}\partial x_{2}^{2}}-C_{3}\frac{\partial^{4}w}{\partial x_{2}^{4}}\\ +\left(C_{9}+C_{11}\right)\frac{\partial^{3}\phi_{1}}{\partial x_{1}\partial x_{2}^{2}}+C_{10}\frac{\partial^{3}\phi_{2}}{\partial x_{2}^{3}}+C_{12}\frac{\partial^{3}\phi_{2}}{\partial x_{1}^{2}\partial x_{2}}-J_{4}\frac{\partial \phi_{2}}{\partial x_{2}}=0\\ L_{3}(\Phi ,w,\phi_{1},\phi_{2})\equiv B_{1}h\frac{\partial^{4}\Phi }{\partial x_{1}^{4}}+\left(2B_{2}+B_{5}\right)h\frac{\partial^{4}\Phi }{\partial x_{1}^{2}\partial x_{2}^{2}}+B_{1}h\frac{\partial^{4}\Phi }{\partial x_{2}^{4}}+B_{9}\frac{\partial^{3}w}{\partial x_{1}^{3}}+(B_{7}+B_{11})\frac{\partial^{3}w}{\partial x_{1}\partial x_{2}^{2}}\;\\ +\frac{1}{r}\frac{\partial^{2}w}{\partial x_{1}^{2}}-B_{4}\frac{\partial^{4}w}{\partial x_{1}^{4}}-\left(2B_{3}-B_{6}\right)\frac{\partial^{4}w}{\partial x_{1}^{2}\partial x_{2}^{2}}-B_{4}\frac{\partial^{4}w}{\partial x_{2}^{4}}+(B_{10}+B_{12})\frac{\partial^{3}\phi_{1}}{\partial x_{1}^{2}\partial x_{2}}+B_{8}\frac{\partial^{3}\phi_{2}}{\partial x_{2}^{3}}=0\\ L_{4}(\Phi ,w,\phi_{1},\phi_{2})\equiv \frac{h}{r}\frac{\partial^{2}\Phi }{\partial x_{1}^{2}}-\frac{Pr}{2}\frac{\partial^{2}w}{\partial x_{1}^{2}}-Pr\frac{\partial^{2}w}{\partial x_{2}^{2}}+J_{3}\frac{\partial \phi_{1}}{\partial x_{1}}+J_{4}\frac{\partial \phi_{2}}{\partial x_{2}}=0. \end{array}$$
where $C_{i},B_{i},J_{l}(i=1,2,\ldots ,12,\;l=3,4)$ are given in Appendix A.

## 4. Solution Procedure

The FG-coated sandwich cylindrical shells are assumed to be clamped at the edges, so the boundary conditions for $x_{1}=0$ and $x_{1}=L$ are as follows [32,34,35,36,37,38]:(19)$$w=0,\;v=0,\;\phi_{1}=0,\;\phi_{2}=0,\;at\;x_{1}=0,L$$

The approximation functions are expressed as [32]:(20)$$\begin{array}{l} \Phi =A_{1}\sin^{2}(k_{1}x_{1})\sin(k_{2}x_{2}),\;w=A_{2}\sin^{2}(k_{1}x_{1})\sin(k_{2}x_{2}),\\ \phi_{1}=A_{3}\cos(k_{1}x_{1})\sin(k_{1}x_{1})\sin(k_{2}x_{2}),\;\phi_{2}=A_{4}\sin^{2}(k_{1}x_{1})\cos(k_{2}x_{2}) \end{array}$$
where $A_{i}(i=1,2,\ldots ,4)$ are amplitudes $k_{1}=\frac{m\pi }{L}$ and $k_{2}=\frac{n}{r}$, in which (m,n) are the longitudinal and circumferential wave numbers, respectively.

The Galerkin method is applied to the system of Equation (18):(21)$$\int_{0}^{2\pi r}\int_{0}^{L}L_{i}(\Phi ,w,\phi_{1},\phi_{2})\sin^{2}(k_{1}x_{1})\sin(k_{2}x_{2})dx_{1}dx_{2}=0\;(i=1,2,\ldots ,4)$$

Substituting (20) into Equation (21), after integration and some mathematical operations, we obtain the following expression for the dimensionless hydrostatic buckling pressure (DHBP) of the FG-coated sandwich cylindrical shells with homogeneous isotropic cores (ceramic- or metal-rich) under clamped boundary conditions based on the FOST:(22)$$P_{1Hbuc}^{ST}=\frac{1}{Y_{c}t_{H}}\frac{u_{22}u_{11}-u_{12}u_{21}}{u_{11}}$$
where $P_{Hbuc}^{ST}=P_{1Hbuc}^{ST}Y_{c}$ is the dimensional hydrostatic buckling pressure (in Pa) within the ST and the following definitions apply:(23)$$\begin{array}{l} u_{11}=z_{21}-\frac{z_{11}z_{23}}{z_{13}},\;u_{12}=z_{22}-\frac{z_{12}z_{23}}{z_{13}},\;u_{21}=z_{31}-\frac{z_{11}z_{33}}{z_{13}},\;u_{22}=z_{32}-\frac{z_{12}z_{23}}{z_{13}},\\ z_{11}=t_{21}-\frac{t_{11}t_{24}}{t_{14}},\;z_{12}=\frac{t_{12}t_{24}}{t_{14}}-t_{22},\;z_{13}=t_{23}-\frac{t_{24}t_{13}}{t_{14}},\;z_{21}=t_{31}-\frac{t_{11}t_{34}}{t_{14}},\\ z_{22}=\frac{t_{12}t_{34}}{t_{14}}-t_{32},\;z_{23}=t_{33}-\frac{t_{13}t_{34}}{t_{14}},\;z_{31}=t_{41}-\frac{t_{11}t_{44}}{t_{14}},\;z_{32}=\frac{t_{12}t_{44}}{t_{14}},\;z_{33}=t_{43}-\frac{t_{13}t_{44}}{t_{14}}. \end{array}$$
in which
(24)$$\begin{array}{l} t_{11}=k_{1}{}^{2}k_{2}{}^{2}\left(C_{1}-C_{5}\right)h+4k_{1}{}^{4}c_{12}h,\;t_{12}=4k_{1}{}^{4}C_{3}+k_{1}{}^{2}k_{2}{}^{2}\left(C_{4}+C_{6}\right),\;t_{14}=k_{1}{}^{2}k_{2}\left(C_{8}+C_{12}\right),\\ t_{13}=2C_{7}k_{1}{}^{3}+0.5k_{1}k_{2}{}^{2}C_{11}+0.5k_{1}J_{3},\;t_{21}=4k_{1}{}^{2}C_{2}h+k_{1}{}^{2}k_{2}{}^{2}h\left(C_{1}-C_{5}\right),\;\\ t_{22}=0.75C_{3}k_{2}{}^{4}+\left(C_{4}+C_{6}\right)k_{1}{}^{2}k_{2}{}^{2},\;t_{23}=0.5k_{1}k_{2}{}^{2}\left(C_{9}+C_{11}\right),\\ t_{24}=0.75k_{2}J_{4}+0.75k_{2}{}^{3}C_{10}+C_{12}k_{1}{}^{2}k_{2},\;t_{31}=h\left(4k_{1}{}^{4}+0.75k_{2}{}^{4}\right)B_{1}+hk_{1}{}^{2}k_{2}{}^{2}\left(B_{5}+2B_{2}\right),\\ t_{32}=\left(2B_{3}-B_{6}\right)k_{1}{}^{2}k_{2}{}^{2}+0.75B_{4}k_{2}{}^{4}+\frac{1}{r}k_{1}{}^{2}+4B_{4}k_{1}{}^{4},\;t_{33}=2B_{9}k_{1}{}^{3}+0.5\left(B_{7}+B_{11}\right)k_{2}{}^{2}k_{1},\\ t_{34}=0.75B_{8}k_{2}{}^{3}+\left(B_{10}+B_{12}\right)k_{1}{}^{2}k_{2},\;t_{41}=\frac{hk_{1}{}^{2}}{r},t_{43}=0.5J_{3}k_{2},\;t_{44}=0.75J_{4}k_{2},\\ t_{H}=0.5ak_{1}^{2}r+0.75k_{2}^{2}r. \end{array}$$

Ignoring the transverse shear strains, the following expression is obtained for the DHBP of the FG-coated sandwich cylindrical shells with homogeneous isotropic cores under clamped boundary conditions based on the CT:(25)$$\begin{array}{l} P_{1Hbuc}^{CT}=\frac{1}{Y_{c}^{\left(2\right)}r(2k_{1}^{2}+3k_{2}^{2})}\{16C_{3}k_{1}^{4}+8k_{1}^{2}k_{2}^{2}(C_{4}+C_{6})+3C_{3}k_{2}^{4}+[\frac{\;4k_{1}^{2}\;}{r\;}-16C_{2}k_{1}^{4}\\ -8k_{1}^{2}k_{2}^{2}(C_{1}-C_{5})-3C_{2}k_{2}^{4}]\times \frac{16B_{4}k_{1}^{4}+4k_{1}^{2}k_{2}^{2}(2B_{3}-B_{6})+3B_{4}k_{2}^{4}+4k_{1}^{2}/r}{16B_{1}k_{1}^{4}+4k_{1}^{2}k_{2}^{2}(2B_{2}+B_{5})+3B_{1}k_{2}^{4}}\} \end{array}$$
where $P_{Hbuc}^{CT}=P_{1Hbuc}^{CT}Y_{c}$ is the DHBP within CT.

The minimum values of the DHBP of the FG-coated cylinders with clamped edges based on the FOST and CT are found by minimizing according to the *m* and *n* wave numbers.

## 5. Numerical Results and Discussion

This section consists of two subsections. The accuracy of the analytical formulas is confirmed under the first subheading. Under the second subheading, the effects of the FG coatings on the DHBP are examined in detail within the framework of the FOST and CT by performing original analyses and providing comments. In all computations, the values in parentheses are the circumferential wave numbers (*n*_cr_) corresponding to the minimum values of the dimensionless hydrostatic buckling pressure (DHBP). Furthermore, it has been determined that the number of longitudinal waves corresponding to the minimum value of the hydrostatic buckling pressure is equal to one (*m* = 1).

### 5.1. Comparison

Table 1 presents the magnitudes of the DHBP of the cylindrical shells consisting of homogeneous isotropic material under clamped boundary conditions. Our calculations are made according to Equation (25), and the following material properties and geometric characteristics of the single-layer cylindrical shells are: $Y_{m}=2\times 10^{11}\mathrm{Pa},\;\nu_{m}=0.3$, L/r=2, r/h=100. The $P_{1Hbuc}^{CT}$ values for the clamped boundary conditions are taken from Singer et al. [39]. As can be seen from Table 1, our results seem to be in agreement with the results obtained in the study of Singer et al. [39].

Table 2 presents the magnitudes of the hydrostatic buckling pressure (in kPa) of the homogeneous isotropic cylindrical shells under clamped boundary conditions. Our calculations are made according to Equation (25), and the following material properties and geometric characteristics of the single-layer cylindrical shells are used: $Y_{m}=5.455\times 10^{10}\mathrm{Pa},\;\nu_{m}=0.3$, L=1,2,3 m, r=0.5 m. The $P_{Hbuc}^{CT}(kPa)$ values in the second and third columns are taken from Table 2 and Table 3, presented in ref. [40]. Table 2 shows that our results are in agreement with those obtained in ref. [40].

### 5.2. Novel Applications

In numerical analysis, cylindrical shells with two kinds of functionally graded coatings, cylindrical shells with two kinds of homogenous coatings and two kinds of single-layer cylindrical shells are used (see Figure 2, Figure 4 and Figure 5). The FG coatings are composed of a mixture of silicon nitride (Si_3_N_4_) and stainless steel (SUS304), forming two kinds of sandwich cylindrical shells, designated FG_1_/Si_3_N_4_/FG_1_ and FG_2_/SUS304/FG_2_ or FG_1_/C/FG_1_ and FG_2_/M/FG_2_, respectively (Figure 1). In addition, metal (SUS304)- and ceramic (Si_3_N_4_)-coated sandwich cylindrical shells are designated as M/C/M and C/M/C, respectively (Figure 4). In addition, single-layer cylindrical shells made of ceramic (Si_3_N_4_) and metal (SUS304) are designed and used for comparisons (Figure 5). In all calculations, the ratio of core thickness to coating thickness is indicated by the symbol: $\eta =h_{core}/h_{coat}$. The shear stress shape functions are as follows:$${\bar{f}}_{i}\left({\bar{x}}_{3}\right)=\frac{df_{i}\left({\bar{x}}_{3}\right)}{d{\bar{x}}_{3}}=\cosh({\bar{x}}_{3})-\cosh(1/2)$$

The properties of the FGMs are taken from the monograph of Shen [36]. The Young’s moduli and Poisson’s ratios of the FG coatings as a function of temperature and their values are presented as follows, when *T* = 300 K:$$\begin{array}{l} E_{Si_{3}N_{4}}=3.4843\times 10^{11}(1-3.07\times 10^{-4}T+2.16\times 10^{-7}T^{2}-8.946\times 10^{-11}T^{3})=322.271(\mathrm{Gpa})\\ E_{Sus304}=2.0104\times 10^{11}(1+3.079\times 10^{-4}T-6.534\times 10^{-7}T^{2})=207.788(\mathrm{GPa})\\ \nu_{Sus304}=0.3262(1-2.002\times 10^{-7}T+3.797\times 10^{-7}T^{2})=0.317756,\;\nu_{Si_{3}N_{4}}=0.24 \end{array}$$

The distribution of the magnitudes of DHBP or $P_{1Hbuc}^{CT}$ and $P_{1Hbuc}^{ST}$ for the M/C/M, FG_1_/C/FG_1_, C/M/C and FG_2_/M/FG_2_ sandwich, ceramic and metal single-layer cylindrical shells against r/h are tabulated in Table 3 with r/L=2, η=0.25 and d=1. The $P_{1Hbuc}^{CT}$ and $P_{1Hbuc}^{ST}$ values for the cylindrical shells covered by the FG_1_ and FG_2_ coatings decrease, while the number of circumferential waves increases depending on the increase in the r/h. When the $P_{1Hbuc}^{ST}$ of the FG_1_- and FG_2_-coated sandwich cylinders are compared with the metal- and ceramic-coated homogeneous sandwich cylinders in the framework of the ST, the effects of the FG_1_ and FG_2_ coatings on the dimensionless hydrostatic buckling pressure reduce from (+18.11%) to (+14.29%) and from (−17.7%) to (−13.5%), respectively, as the r/h increment increases from 20 to 50. As the FG_1_- and FG_2_-coated sandwich shells are compared with pure ceramic and pure metal cylindrical shells in the framework of the ST, respectively, the effects of the FG_1_ and FG_2_ coatings on the $P_{1Hbuc}^{ST}$ increase from (−15.72%) to (−16.76%), and from (+17.57%) to (+21.55), respectively, as the r/h ratio increases from 20 to 50. The most significant effect of the transverse shear strains on the DHBP of the FG_1_- and FG_2_-coated sandwich cylindrical shells occurs with 18.34% of the shell covered by the FG_2_ coating at r/h = 20 and decreases by up to 2.84% when r/h = 50. In the shell covered by the FG_1_ coating, this effect is lower than in the FG_2_-coated sandwich shell with the metal core, decreasing from 8.46% to 1.39% as the r/h ratio increases from 20 to 50. Although these influences are evident at small values of r/h in pure ceramic and pure metal shells, they are reduced from 11.4% to 1.73% and from 12.68% to 1.72%, respectively, when r/h increases from 20 to 50.

The distribution of the magnitudes of $P_{1Hbuc}^{CT}$ and $P_{1Hbuc}^{ST}$ for the M/C/M, FG_1_/C/FG_1_, C/M/C and FG_2_/M/FG_2_ sandwich cylindrical shells versus the η are shown in Table 4. The following data and volume fraction index are used: L/r=0.5, r/h=25 and d=1. The magnitudes of $P_{1Hbuc}^{CT}$ and $P_{1Hbuc}^{ST}$ for the three-layered cylinders with ceramic cores increase, while they decrease for the three-layered cylinders with the metal cores, as the η increases. The circumferential wave number corresponding to the DHBP increases with the increase in η. When the $P_{1Hbuc}^{ST}$ of the FG_1_- and FG_2_-coated sandwich cylinders are compared with those of the M/C/M and C/M/C shells, the respective effect on the $P_{1Hbuc}^{ST}$ decreases from (+13.94%) to (+7.86%) for the FG_1_ coating and, although it shows disorder, from (−12.72%) to (−9.42%) for the FG_2_ coating as the η ratio increases from 2 to 8. Furthermore, when the FG_1_- and FG_2_-coated sandwich shells are compared with the pure ceramic and metal single-layer shells, the respective effect on the $P_{1Hbuc}^{ST}$ decreases from (−18.07%) to (−8.98%) for the FG_1_ coating and from (+23.66%) to (+11.52%) for the FG_2_ coating as the η increases from 2 to 8. The most significant effect of the transverse shear strains on the DHBP of the FG_1_ and FG_2_-coated sandwich cylindrical shells occurs at 33.4% in the FG_2_-coated sandwich shell with the metal core at η = 8, and that effect is 18.53% when η = 2. In the FG_1_-coated shell, this effect is lower than in the FG_2_-coated shell, reducing from 8.51% to 8.13% as the η ratio increment from 2 to 8.

The variations in the magnitudes of $P_{1Hbuc}^{CT}$ and $P_{1Hbuc}^{ST}$ for the FG_1_- and FG_2_-coated sandwich cylindrical shells against the d are presented in Table 5. The following data are used: L/r=0.5, r/h=25, η=0.25 and d=1. The magnitudes of $P_{1Hbuc}^{CT}$ and $P_{1Hbuc}^{ST}$ for the FG_1_ kind sandwich cylindrical shells decrease, while they increase for the FG_2_ sandwich cylindrical shells, as the volume fraction index increases. Within the framework of these data, the circumferential wave numbers are independent of the change in d. When the FG_1_- and FG_2_-coated cylinders are compared with the pure ceramic and pure metal single-layer cylinders, the respective effect on the $P_{1Hbuc}^{ST}$ decreases from (−11.79%) to (−4.26%) for the FG_1_ coatings, but increases from (+5.16%) to (+11.57%) for the FG_2_ coatings as the d increases from 0.5 to 2. It is thus revealed that the effect of material heterogeneity on the DGBP decreases significantly with the increase of the d ratio from 0.5 to 2 in both kinds of FG coating. In addition, the coating with the greatest effect on the DHBP is the FG_2_ coating, when compared with the single-layer shells. When the values of the dimensionless hydrostatic buckling pressure of the FG_1_ and FG_2_-coated sandwich cylindrical shells are compared, the values of the DHBP are lower in the ST than in the CT. The most significant effect of the transverse shear strains on the DHBP occurs with 18.78% in FG_2_-coated sandwich shell at d = 2. In the FG_1_-coated sandwich shell, it is lower than in the FG_2_-coated sandwich cylindrical shell, decreasing from 5.96% to 4.34% as the d index increases from 0.5 to 2.

## 6. Conclusions

In this study, the buckling of FG-coated sandwich cylindrical shells was investigated. The most important aspect of this study is the solution of the buckling problem of clamped FG-coated sandwich cylindrical shells subjected to hydrostatic pressure by determining a new approximation function in the framework of the FOST. The basic equations were derived based on the Donnell shell theory, and new analytical expressions for the hydrostatic buckling pressure under clamped boundary conditions were found within the FOST and CT by applying Galerkin’s procedure. Finally, the findings of the present study were verified by comparing with those presented in the literature, and the effects of the FG profiles, shear stresses, volume fractions and shell characteristics on the DHBP were examined in detail.

Numerical analyses and comments revealed the following generalizations:

The $P_{1Hbuc}^{CT}$ and $P_{1Hbuc}^{ST}$ values for the cylindrical shells covered by the FG_1_ and FG_2_ coatings decrease, while the number of circumferential waves increases depending on the increase in the r/h.As the $P_{1Hbuc}^{ST}$ of the FG_1_- and FG_2_-coated sandwich cylinders are compared with the metal- and ceramic-coated homogeneous sandwich cylinders in the framework of the FOST, the influence of the FG_1_ and FG_2_ coatings on the dimensionless hydrostatic buckling pressure decreases as the r/h increases.As the FG_1_ and FG_2_-coated sandwich shells are compared with pure ceramic and pure metal cylindrical shells in the framework of the ST, the effect of the FG_1_ and FG_2_ coatings on the $P_{1Hbuc}^{ST}$ increases as the r/h increases.The most significant effect of the transverse shear strains on the DHBP of the FG_1_- and FG_2_-coated sandwich cylindrical shells occurs in the shell covered by the FG_2_ coating at r/h = 20.The magnitudes of $P_{1Hbuc}^{CT}$ and $P_{1Hbuc}^{ST}$ for the FG_1_ sandwich cylindrical shells decrease, while they increase for the FG_2_ sandwich cylindrical shells, as the volume fraction index increases.When FG_1_- and FG_2_-coated shells are compared with the pure ceramic and pure metal single-layer cylinders, respectively, the effect of the FG_1_ coating on the $P_{1Hbuc}^{ST}$ decreases, whereas the influence of the FG_2_ coating on the $P_{1Hbuc}^{ST}$ increases, as the d increases.The most significant effect of the transverse shear strains on the DHBP occurs in FG_2_-coated sandwich shell at d = 2.As the FG_1_- and FG_2_-coated sandwich cylinders are compared with the pure ceramic and metal single-layer cylinders, the influence of FG_1_ and FG_2_ coatings on the $P_{1Hbuc}^{ST}$ decreases as the η increases.

## Figures and Tables

**Figure 1 materials-15-08680-f001:**
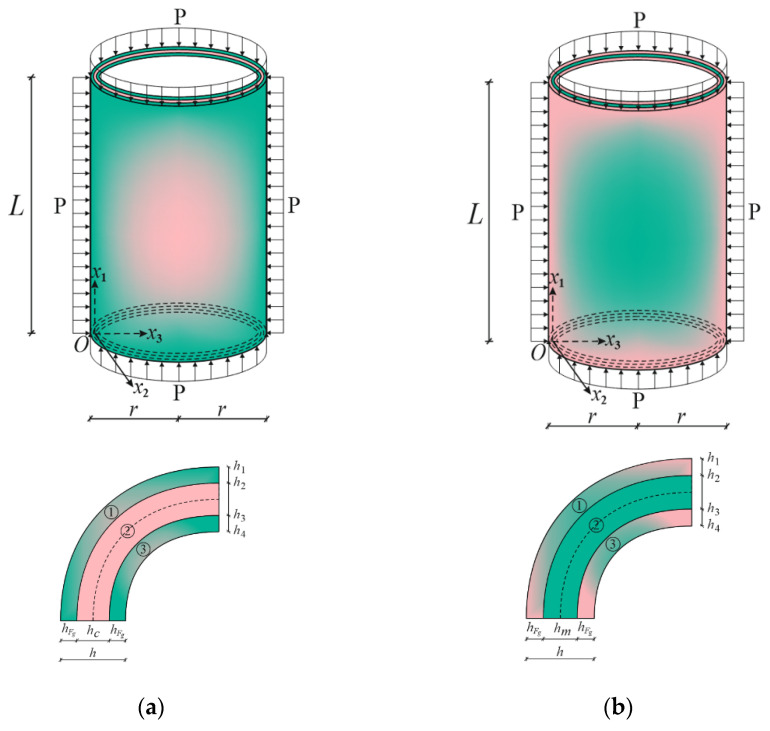
(**a**) FG_1_/C/FG_1_ and (**b**) FG_2_/M/FG_2_ sandwich cylindrical shells under hydrostatic pressure and their cross sections.

**Figure 2 materials-15-08680-f002:**
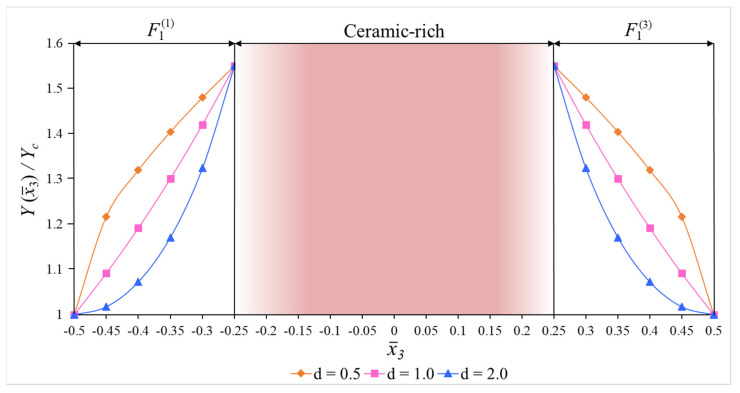
The variations in the dimensionless Young moduli of FG_1_/C/FG_1_ cylindrical shells.

**Figure 3 materials-15-08680-f003:**
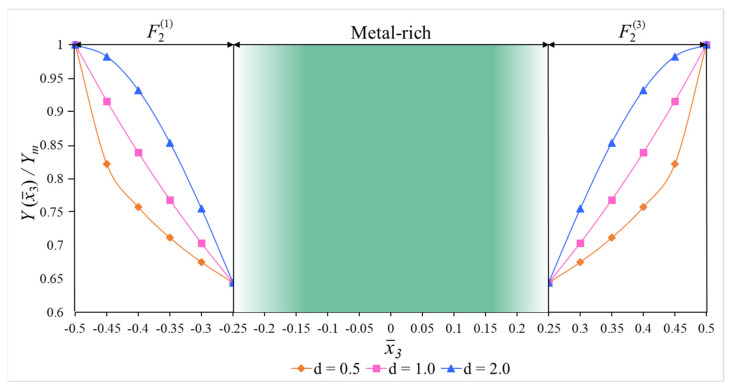
The variations in dimensionless Young moduli of the FG_2_/M/FG_2_ cylindrical shells.

**Figure 4 materials-15-08680-f004:**
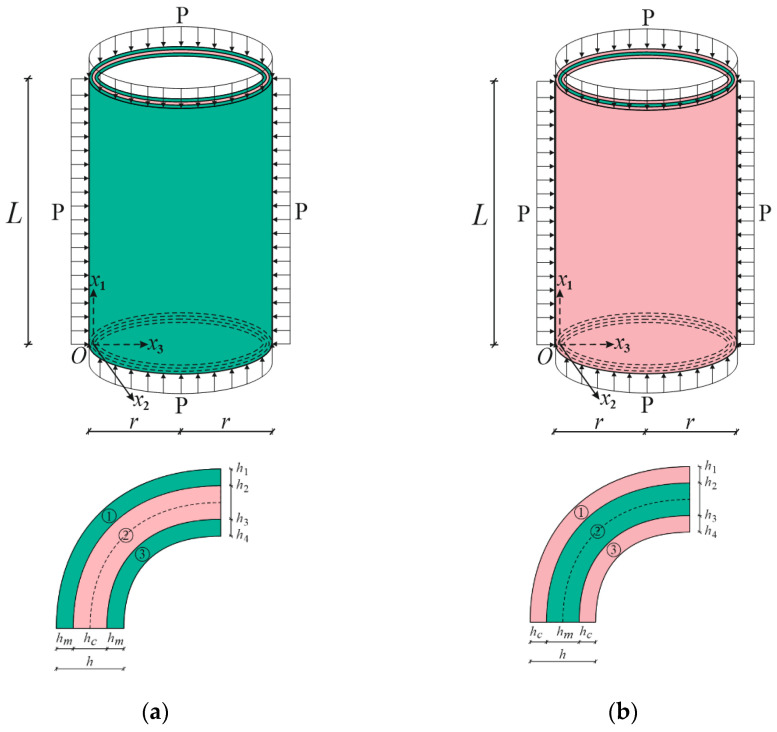
(**a**) M/C/M and (**b**) C/M/C cylindrical shells under hydrostatic pressure and their cross sections.

**Figure 5 materials-15-08680-f005:**
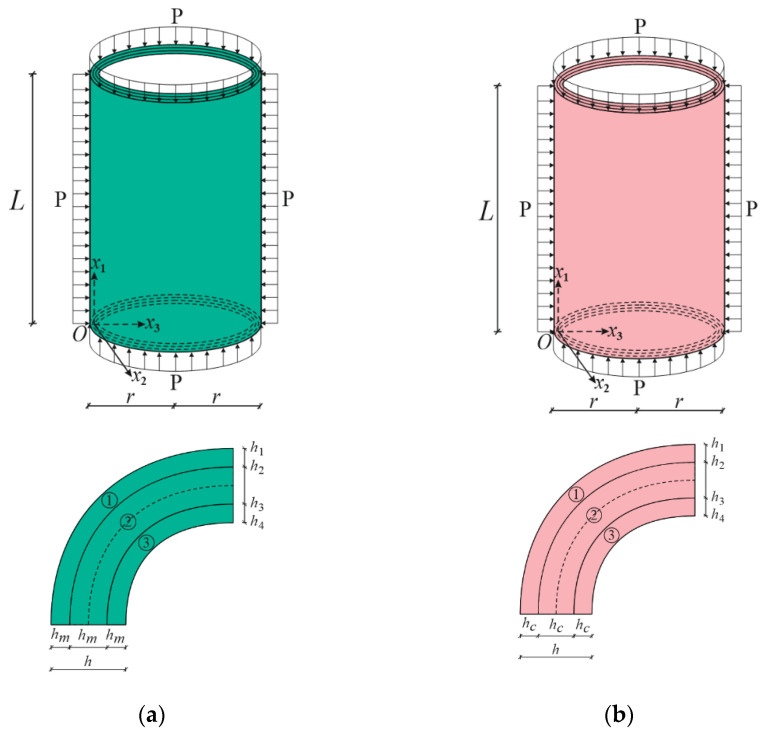
(**a**) Pure metal and (**b**) pure ceramic cylindrical shells under hydrostatic pressure and their cross sections.

**Table 1 materials-15-08680-t001:** Comparison of $P_{1Hbuc}^{CT}$ for single-layer isotropic cylindrical shells with clamped edges.

	$P_{1Hbuc}^{CT}\times 10^{6}$, (*n*_cr_)		
*L*/*r*	Singer et al. [39] CC2 Classic	Singer et al. [39] CC1	Present Study
0.5	27.98 (11)	26.32 (11)	27.456 (11)
1	12.89 (9)	11.03 (8)	11.7789 (9)
2	6.52 (7)	5.026 (7)	5.759 (7)

**Table 2 materials-15-08680-t002:** Comparison of $P_{Hbuc}^{CT}(kPa)$ for single-layer cylindrical shells made of homogeneous material with clamped edges.

	$P_{Hbuc}^{CT}(kPa)$, (*n*_cr_)		
*L*	Lopatin and Morozov [40] FEM	Lopatin and Morozov [40] Analytical	Present Study
1	2003.1	1922.4	1885.09 (6)
2	1027.2	994.4	860.14 (4)
3	724.8	754.9	704.82 (4)

**Table 3 materials-15-08680-t003:** Variations in $P_{1Hbuc}^{CT}$, $P_{1Hbuc}^{ST}$ and (*n*_cr_) of various kinds of sandwich and single-layer cylindrical shells under clamped boundary conditions versus the r/h.

	$P_{1Hbuc}^{ST}$$\times 10^{3}$ (*n*_cr_)	$P_{1Hbuc}^{CT}$$\times 10^{3}$ (*n*_cr_)	$P_{1Hbuc}^{ST}$$\times 10^{3}$ (*n*_cr_)	$P_{1Hbuc}^{CT}$$\times 10^{3}$ (*n*_cr_)	$P_{1Hbuc}^{ST}$$\times 10^{3}$ (*n*_cr_)	$P_{1Hbuc}^{CT}$$\times 10^{3}$ (*n*_cr_)
r/h	M/C/M		FG_1_/M/FG_1_		Ceramic	
20	1.552 (8)	1.745 (8)	1.833 (8)	1.988 (8)	2.175 (8)	2.423 (8)
25	0.846 (8)	0.912 (8)	0.986 (8)	1.038 (8)	1.179 (8)	1.264 (8)
30	0.513 (9)	0.541 (8)	0.594 (9)	0.616 (8)	0.713 (8)	0.748 (8)
40	0.231 (9)	0.239 (9)	0.276 (8)	0.271 (8)	0.320 (9)	0.329 (9)
50	0.126 (10)	0.128 (10)	0.144 (10)	0.146 (10)	0.173 (9)	0.176 (9)
r/h	C/M/C		FG_2_/M/FG_2_		Metal	
20	2.073 (8)	2.314 (8)	1.706 (8)	2.020 (8)	1.451 (8)	1.635 (8)
25	1.121 (8)	1.204 (8)	0.942 (8)	1.051 (8)	0.789 (8)	0.852 (8)
30	0.676 (8)	0.710 (8)	0.575 (8)	0.620 (8)	0.478 (8)	0.504 (8)
40	0.303 (9)	0.312 (9)	0.260 (9)	0.272 (9)	0.215 (9)	0.221 (9)
50	0.163 (9)	0.166 (9)	0.141 (9)	0.145 (9)	0.116 (9)	0.118 (9)

**Table 4 materials-15-08680-t004:** Variations of $P_{1Hbuc}^{CT}$, $P_{1Hbuc}^{ST}$ and (*n*_cr_) in various kinds of sandwich and monolayer cylindrical shells under clamped boundary conditions versus the η.

	$P_{1Hbuc}^{ST}$$\times 10^{3}$ (*n*_cr_)	$P_{1Hbuc}^{CT}$$\times 10^{3}$ (*n*_cr_)	$P_{1Hbuc}^{ST}$$\times 10^{3}$ (*n*_cr_)	$P_{1Hbuc}^{CT}$$\times 10^{3}$ (*n*_cr_)	$P_{1Hbuc}^{ST}$$\times 10^{3}$ (*n*_cr_)	$P_{1Hbuc}^{CT}$$\times 10^{3}$ (*n*_cr_)
η	M/C/M		FG_1_/C/FG_1_		Ceramic	
2	1.222 (10)	1.377 (10)	1.446 (10)	1.569 (10)	1.717 (10)	1.915 (10)
4	1.320 (10)	1.483 (10)	1.532 (10)	1.655 (10)		
6	1.391 (10)	1.561 (10)	1.581 (10)	1.708 (10)		
8	1.442 (10)	1.616 (10)	1.612 (10)	1.743 (10)		
η	C/M/C		FG_2_/M/FG_2_		Metal	
2	1.639 (10)	1.832 (9)	1.349 (10)	1.599 (9)	1.146 (10)	1.293 (10)
4	1.540 (10)	1.725 (9)	1.242 (10)	1.521 (9)		
6	1.469 (10)	1.648 (9)	1.159 (10)	1.474 (10)		
8	1.418 (10)	1.592 (9)	1.081 (10)	1.442 (10)		

**Table 5 materials-15-08680-t005:** Variations in $P_{1Hbuc}^{ST}$, $P_{1Hbuc}^{CT}$ and (*n*_cr_) of FG_1_ and FG_2_ sandwich, pure metal and pure ceramic cylindrical shells versus the d.

Volume Fraction Index ( d)	FG_1_/C/FG_1_		FG_2_/M/FG_2_	
	$P_{1Hbuc}^{ST}$$\times 10^{3}$ (*n*_cr_)	$P_{1Hbuc}^{CT}$$\times 10^{3}$ (*n*_cr_)	$P_{1Hbuc}^{ST}$$\times 10^{3}$ (*n*_cr_)	$P_{1Hbuc}^{CT}$$\times 10^{3}$ (*n*_cr_)
0.5	1.040 (8)	1.102 (8)	0.896 (8)	0.984 (8)
1	0.986 (8)	1.038 (8)	0.942 (8)	1.051 (8)
2	0.944 (8)	0.985 (8)	0.937 (8)	1.113 (8)
	Pure ceramic		Pure metal	
	1.179 (8)	1.264 (8)	0.789 (8)	0.852 (8)

## Data Availability

No data were reported in this study.

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
