# Peer review of "Buckling Behavior of Sandwich Cylindrical Shells Covered by Functionally Graded Coatings with Clamped Boundary Conditions under Hydrostatic Pressure"

_materials, 2022, doi:10.3390/ma15238680_

Round 1
Reviewer 1 Report
Please find attached file.

Author Response
Report of Reviewer 1 attached

Reviewer 2 Report
"Buckling Behavior of Sandwich Cylindrical Shells Covered by Functionally Graded Coatings with Clamped Boundary Conditions under Hydrostatic Pressure", the article is interesting. A few observations are given below,
1) The abstract is not clear. An abstract is a short summary of your completed research. It is intended to describe your work without going into great detail. Abstracts should be self-contained and concise, explaining your work as briefly and clearly as possible.
The abstract is too short, Objectives and aims are missing in the abstract
2) For readers to quickly catch the contribution of this work, it would be better to highlight major difficulties and challenges, and the authors' original achievements to overcome them, in a clearer way in the Introduction section.
3) The results and discussion are not clearly dealt with the outcomes of the proposed work. The authors should explicitly state the novel contribution of this work, and the similarities, and differences between this work with the previous publications in this section.
Author Response
Report of Reviewer 2 attached

Reviewer 3 Report
This study proposed an analytical solution to the buckling problem of functionally graded (FG) sandwich cylindrical shells exposed to clamped boundary conditions and hydrostatic pressure within the framework of shear deformation shell theory (ST). By mathematically modeling FG-coatings, constitutive relations and fundamental equations for sandwich cylinder shells within ST are derived. Using new approximation functions that satisfy clamped boundary conditions, basic equations are solved analytically within the framework of the Donnell shell theory utilizing the Galerkin technique. This paper can be accepted for publication after the author considers some points as follows:
1. The first sentence of the first paragraph in Section 1 should be cited from value documents. Similarly, the second one, "The idea of producing FGM was developed by Japanese materials scientists in 1984 to increase the thermal barriers between indoor and outdoor temperatures using only 10 mm thickness". It is not difficult to find reviews of these conclusions
2. The authors have written the structure of the Introduction section quite logically. Several articles related to FGM materials can be cited further to enrich this section. "Finite element modelling for free vibration response of cracked stiffened FGM plates"; "Bending of Symmetric Sandwich FGM Beams with Shear Connectors"; "Numerical Investigation on Static Bending and Free Vibration Responses of Two-Layer Variable Thickness Plates with Shear Connectors"; "The Third-Order Shear Deformation Theory for Modeling the Static Bending and Dynamic Responses of Piezoelectric Bidirectional Functionally Graded Plates"; "Research on the buckling behavior of functionally graded plates with stiffeners based on the third-order shear deformation theory".
3. Font size in MathType and the text don't seem to be compatible. The author should double-check this point.
4. Since section 2 describes the geometrical dimensions and the rules of the FGM material of the shell, so the section "2. Formulation of problem" should be changed to "2. Material and geometrical model of problem"
5. If possible, the author should plot the change of the volume fraction in the radial direction as described in Equation (1) of the shell so that the reader can visualize it more intuitively.
6. Equation (13) should be cited from reputable scientific works.
7. In the problem of comparing results, document number [31] is quite outdated and seems to use the same analytical method as the method used by the author in this problem. If possible, the author should compare with numerical methods to make your conclusion more convincing.
8. In the first problem description for "5.2. Novel Applications", the mechanical properties of FGM materials should be described in a table with citations.
9. For one case, could the author draw the deformed shape of the shell structure?
10. Some grammatical and typo mistakes were found in the article; the authors should review the entire article.
Author Response
Report of Reviewer 3 attached

Round 2
Reviewer 1 Report
Please find attached file

Author Response
Explonation to Reviewer 1 is attaced

Reviewer 3 Report
All anwers are clear
Author Response
Explonation to Reviewer 3 is attaced

Round 3
Reviewer 1 Report
I satisfied the reviewer responses. So, the manuscript can be recommended for publication in Materials.